# Top-funded digital health companies offering lifestyle interventions for dementia prevention: Company overview and evidence analysis

Rasita Vinay[1,2,3]⊙*, Jonas Probst[2]⊙, Panitda Huynh[2], Mathias Schlögl[4], Tobias Kowatsch[2,3,5]‡, Marcia Nißen[2,3]

1 Institute of Biomedical Ethics and History of Medicine, University of Zurich, Zurich, Switzerland, 2 School of Medicine, University of St. Gallen, St. Gallen, Switzerland, 3 Centre for Digital Health Interventions, Department of Management, Technology and Economics, ETH Zurich, Zurich, Switzerland, 4 Clinic Barmelweid, Aargau, Switzerland, 5 Institute for Implementation Science in Health Care, University of Zurich, Zurich, Switzerland

⊙ These authors contributed equally to this article and share first authorship.
‡ TK and MN also contributed equally to this article and share last authorship.
* rvinay@ethz.ch

## Abstract

### Background and objective

Dementia prevention has been recognized as a top priority by public health authorities due to the lack of a reversible cure. In this regard, digital dementia-preventive lifestyle services (DDLS) emerge as potentially pivotal services, aiming to address modifiable risk factors on a large scale. This study aims to identify the top-funded companies offering DDLS and evaluate their clinical evidence to gain insights into the international service landscape.

### Methods

A systematic screening of two financial databases (Pitchbook and Crunchbase) was conducted. Corresponding published clinical evidence was collected through a systematic literature review and analyzed regarding study purpose, results, quality of results, and level of clinical evidence.

### Findings

The ten top-funded companies offering DDLS received a total funding of EUR 128.52 million, of which three companies collected more than 75%. Clinical evidence was limited due to only nine eligible publications, small clinical subject groups, the absence of longitudinal study designs, and no direct evidence of dementia prevention.

### Conclusion

Our study shows that the level of funding received by companies does not reflect the clinical effectiveness of DDLS. The study serves as an initial step toward understanding how DDLS are currently evaluated in today's market and highlights the need for a more rigorous evaluation of DDLS effectiveness.

**Data availability statement:** All relevant data are within the manuscript and its Supporting Information files.

**Funding:** The author(s) received no specific funding for this work.

**Competing interests:** RV, PH, TK, and MN are affiliated with the Centre for Digital Health Interventions (CDHI), a joint initiative of the Institute for Implementation Science in Health Care, University of Zurich, the Department of Management, Technology, and Economics at the ETH Zurich, and the institute of Technology Management and School of Medicine at the University of St. Gallen. CDHI is funded in part by CSS, a Swiss health insurer, Mavie Next, an Austrian healthcare provider and MTIP, a Swiss investor company. TK is also a co-founder of Pathmate Technologies, a university spin-off company that creates and delivers digital clinical pathways. However, neither CSS nor Pathmate Technologies, Mavie Next, or MTIP was involved in this research. This does not alter our adherence to PLOS ONE policies on sharing data and materials. All other authors declare no conflict of interest.

## Introduction

Dementia is an umbrella term covering a range of neurodegenerative syndromes that cause cognitive decline severe enough to interfere with daily life [1]. While it is common for individuals to experience mixed dementia [2], Alzheimer's Disease (AD) is the most common cause of dementia [3]. Other prevalent forms include Lewy body dementia, vascular dementia, and frontotemporal disorders [1,3]. The wide spectrum of symptoms associated with dementia depend on its form and include, amongst others, memory loss, confusion, speaking difficulties, disorientation in familiar surroundings and hallucinations [4]. Especially in its initial phases, the diagnosis of dementia can pose challenges and become difficult to distinguish from typical age-related changes [5,6].

According to the World Health Organization (WHO), more than 55 million people worldwide are currently affected by dementia, and projections indicate a rise to 78 million in 2030 and a staggering 139 million people in 2050 [7]. With a global prevalence of 6.9% in the age group above 65 years, dementia has become one of the leading causes of care dependence in old age and the seventh leading cause of global deaths [7].

Since there is no reversible cure for dementia, prioritizing prevention strategies becomes a public health priority. Although dementia risk is strongly correlated with age, studies indicate that lifestyle, particularly from midlife onwards, significantly influences individuals' susceptibility to developing the condition later in life [8]. Modifiable risk factors that influence dementia risk include, among others, physical activity, diet, and smoking [9,10]. It is estimated that 45% of all dementia can be prevented through the targeted change of these modifiable risk factors during midlife and older age [11]. Due to the long preclinical phase of dementia and the challenges associated with directly measuring prevention success within typical study durations [11,12], many lifestyle-based prevention trials rely on proxies such as cognitive performance or memory function [13], or lifestyle adherence to assess short-term effectiveness [14], rather than demonstrating direct reductions in dementia incidence [11–14]. The WHO confirms the important role of prevention by making the capacity improvement of healthcare professionals for the proactive management of modifiable risk factors a main target in its global action plan against dementia [15]. The social and economic effects of dementia are severe, where a person with dementia (PWD) is significantly more likely to be hospitalized and have a substantially higher average length of stay in hospitals [16]. In 2019, dementia incurred an estimated global cost of over USD 1.3 trillion, translating to approximately USD 24,000 per PWD [7]. Informal care provided by family members, friends, and neighbors makes up almost 50% of total dementia-associated costs [7]. Caregivers are often referred to as "invisible second patients", as they often do not receive adequate support and care for the tasks which they are required to fulfill, negatively impacting their own health needs and well-being [17,18]. Due to this, caregivers have a higher likelihood of experiencing depression and anxiety [19], along with an increased risk of

developing cardiovascular diseases [20] due to their caregiving duties. To address the social and economic ramifications of dementia, modern societies must strengthen their capabilities of dementia prevention by leveraging scalable and cost-effective approaches. Amid the global demographic shift and increasing labor shortage in healthcare [21], digital health interventions (DHIs) could play a central role in prevention and in delivering scalable, personalized, and evidence-based interventions [22].

DHIs are part of the broader concept of Digital Health and pose new opportunities to bridge the gap in care access and quality in health systems [23–25]. They offer various benefits to all actors in the system [25] and an increasing body of evidence in various disease domains suggests positive effects of DHIs on costs and health outcomes [22].

Digital dementia-preventive lifestyle services (DDLS) build on the advantages of DHIs and have the potential to reach larger populations, potentially also those in underserved or rural areas, and to provide continuous support independent of the financial, geographical, or time constraints of traditional face-to-face counselling. Against the background of upcoming demand for DDLS and an increasing number of companies in the field of DHIs, this study aims to provide an initial mapping of top-funded companies and to assess the current state of published clinical evidence linked to their services by answering the following research questions:

RQ1. What are the globally top-funded digital dementia-preventive lifestyle services?

RQ2. What is the clinical evidence of the identified solutions?

## Materials and methods

In this section, we present the methodologies of two studies we conducted against the research questions. Following the procedure described in Safavi et al. (2019), study 1 provides an identification of DDLS companies through market screening, and study 2 provides an evidence analysis of published clinical studies by identified companies. This methodological approach provides us with the current state of peer-reviewed clinical validation of the top funded DDLS companies [26] To ensure reliability and credibility, the research team incorporated a peer debriefing method, as common for qualitative research [27].

### Study 1: Company overview

**Search strategy.** To ascertain the top funded companies offering DDLS, this study combined data extraction from two leading financial databases, Pitchbook and Crunchbase. This multi-pronged approach ensured a broad capture of the landscape, identifying enterprises with significant funding aimed at dementia prevention through digital lifestyle interventions. The search strategy was refined through an iterative process among the co-authors, with keywords in three categories: "Verticals, methods, and industries", "Dementia", and "Management and prevention".

Since the search strategy aimed to capture companies offering DDLS targeting all-cause dementias, including AD, vascular dementia, Lewy body dementia, and frontotemporal dementia, the search terms used in both Pitchbook and Crunchbase included not only the term "dementia" but also related terms such as "Alzheimer", "cognitive decline", and "memory loss" to account for the broad range of conditions that fall under the dementia umbrella (see S1 and S2 Tables).

Due to limited keyword search masks in Crunchbase, "Dementia" and "Management and prevention" categories were merged. S1 and S2 Tables provide the selected search strategies and keywords used in both databases, which defer due to differences in the search functions of the two databases, where Crunchbase did not utilize OR/AND operators, and only predefined industries could be selected.

To supplement and further validate the searches on Crunchbase and Pitchbook, we conducted an additional exploratory web search using Google, aiming to identify any highly visible companies potentially not (yet) captured in the financial databases. All companies, regardless of their source of identification, were screened and selected using the same predefined eligibility criteria, ensuring consistency in the final sample. The full selection process is illustrated in the 'sample characteristics section' below.

**Inclusion and exclusion criteria.** The inclusion criteria were stringently designed to focus on digital health technologies directly targeting patients or consumers with interventions capable of potentially modifying lifestyle factors associated with dementia risk. Essential for inclusion were technologies that demonstrated a clear application towards dementia prevention, articulated through their digital solutions. Exclusion criteria were carefully applied to omit companies not directly targeting dementia risk, lacking in necessary detail, lacking funding information, or not providing solutions in English, ensuring a focus on globally applicable and accessible services.

**Selection process.** An intricate screening process ensued, beginning with the elimination of duplicates and a thorough review of database entries and company websites. Each company was evaluated against the inclusion and exclusion criteria by a dedicated researcher (JP), with a subsequent independent review by a second researcher to ensure thoroughness and reliability (MN). Disagreements were solved through discussion and the interrater reliability was assessed through the calculation of the Cohen's kappa coefficient. This two-tiered review process was augmented by expert feedback, soliciting insights from academicians and industry specialists in dementia care and prevention. Experts were invited to assess the preliminary list and suggest additional companies, further enriching the dataset.

**Data collection.** Based on previous research [26], the following data points were collected via Pitchbook (as available by January 30, 2023): Year founded, headquarter location, total amount raised, last financing size, last financing date, last financing type, years of funding rounds, number of financing rounds, number of investors and number of employees. Considered financing rounds included all deal types (Angel, Seed, Early-Stage VC, Later Stage VC, Equity Crowdfunding, PE Growth/Expansion, Corporate, Joint Venture, M&A) and were only included if completed by January 30, 2023. In case of unavailable funding information, corresponding information was retrieved from the Crunchbase database, or further from publicly available news articles to identify the last financing size as indicator for the overall funding amount.

**Sample characteristics.** The search iteration yielded a cumulative outcome of 605 total results (Pitchbook: 341, Crunchbase: 262, Google: 2), and 16 duplicates were removed. The screening of the database company descriptions resulted in further exclusion of 573 additional companies. Out of the remaining 16 companies, funding information could not be obtained for one. In the collaborative coding process according to the pre-defined set of inclusion and exclusion criteria, six companies were excluded. The final list of nine companies were reviewed by four experts who suggested 15 additional companies for review. On a scale of one (no expertise) to five (extremely high expertise), the experts assessed their market expertise at an average score of 2.6. Of those 15 expert-suggested companies, two were already included from the database results, and 12 were excluded in the collaborative coding process. This process provided us with a list of nine companies from the database search, and one company through expert feedback, resulting in a total of 10 companies. The detailed flow of company inclusion and exclusion, including the number of excluded companies per reason, is illustrated in Fig 1. The Cohens' kappa coefficient prior to expert validation was $k_1 = 0.857$ (93% agreement), and after expert validation was $k_E = 0.66$ (86.6% agreement). These values align with a strong and moderate level of agreement [28], consequently establishing the reliability of the results.

**Data analysis.** Company data was extracted and analyzed with descriptive statistics by one researcher (JP).

## Study 2: Evidence analysis

**Search strategy.** Peer-reviewed publications were identified by searching Google Scholar and PubMed for the company name and by retrieving study references on corresponding company websites as available by April 18, 2023. Please note: Google Scholar search results primarily cover titles, abstracts, author lists, and (where accessible) the full

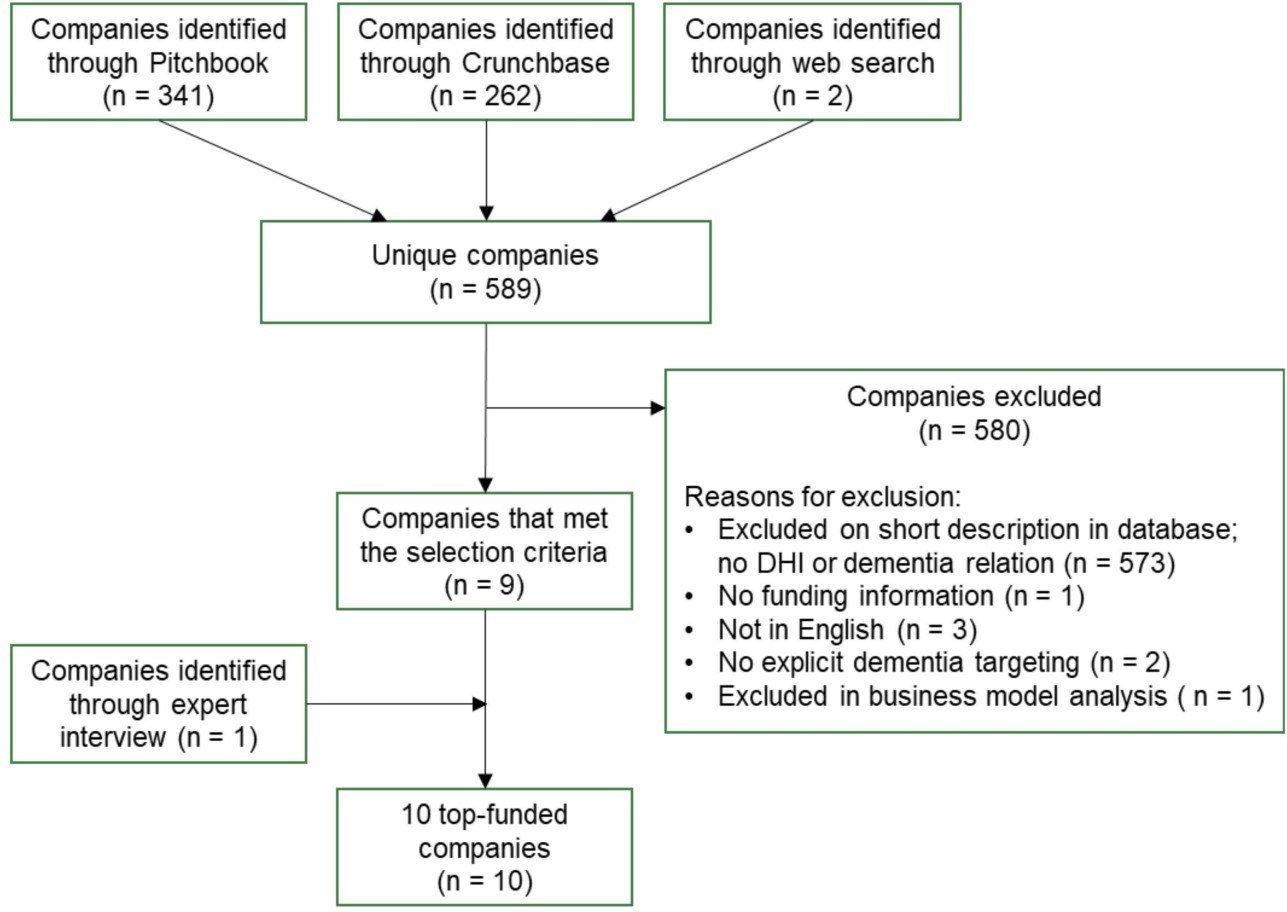

**Fig 1. Flow diagram for the included DHIs for the prevention of dementia in the systematic market analysis.**

text of open access publications. In PubMed, searches cover titles, abstracts, author lists, author affiliations as well as grant and funding information (for studies published from 2014 onwards), but not full text searches.

In case the name of the solution differed from the company name, databases were searched for both names using the OR boolean operator. If necessary (due to an unmanageable number of search results), Google Scholar searches were further limited through the publishing date after founding year of the corresponding company and/or with the keyword "Dementia" (using the AND operator), as additional search requirements. Given that our study specifically examines dementia-preventive lifestyle interventions, the term "Dementia" was used to refine search results and maintain alignment with our research focus. This filter was applied selectively and only in cases where broader searches produced results unrelated to dementia prevention.

**Inclusion and exclusion criteria.** Identified studies were sought to be relevant if they were peer-reviewed publications that examine the potential effects of the identified solutions on clinical outcome, cost, or access to care in dementia care or dementia prevention, aligning with the broader impact categories outlined in the market analysis framework by Safavi et al. (2019) [26]. Exclusions were made for studies on non-dementia conditions (e.g., Parkinson's disease), usability or pre-studies with non-risk populations (e.g., healthy college students), non-English publications, protocols, proof-of-concept works, systematic reviews, or commentaries, due to their irrelevance for dementia prevention.

**Selection process.** After the removal of duplicates, title and abstract screening of publications was conducted by one author (JP), and full-text review was conducted by two authors (JP and MN). In case of disagreements, consensus was achieved through discussion. Interrater reliability was again assessed through the calculation of the Cohen's kappa.

**Data collection.** In line with the procedure described in [26], publications were analyzed for evidence level, the number of clinical subjects, the purpose of the study, target condition or risk factor (if specifically targeted), and the demonstrated effect as per U.S. Preventive Services Task Force (USPSTF). USPSTF levels of evidence are: Level 1 (good) with at least one randomized trial, Level 2 (fair) includes non-randomized or well-designed studies, and Level 3 (poor) consists of expert opinions or descriptive studies [26].

The purpose of this categorization was to classify studies as: effectiveness, validation, or other studies. In the coding process, the journal and paper type, trial registration, and demonstrated changes in utilized proxies were retrieved in addition to the data from prior research [26]. Moreover, on a scale of 1 (low) to five (high), subjective quality assessment scores (referring to the relevance for answering the research question), were assigned to each study.

**Sample characteristics.** 1,890 publications were identified from database search. After removing 129 duplicates, 1,784 unique articles underwent title and abstract screening. 1,693 articles were excluded based on the inclusion and exclusion criteria. After full-text review, 82 articles were further excluded, for instance, due to a missing relation to the identified companies or products (n = 23), systematic reviews (n = 19), or missing focus on dementia (n = 13). Nine articles were deemed eligible for evaluation of clinical evidence. The detailed flow of article inclusion and exclusion, including the number of excluded articles per reason, is illustrated in Fig 2. A Cohens' kappa coefficient of $k_0 = 1$ was determined during the full-text review, while the following coding-based analysis led to an initial $k_1 = 0.57$ (77% agreement) and a $k_2 = 1$ (100% agreement) after discussion.

**Data analysis.** Data extraction and analysis was conducted by two authors (JP, MN) and followed a one-cycle coding process based on previously introduced publication-related aspects [26]. In case of disagreement, consensus was reached through discussion and interrater reliability was reported.

## Results

### Top funded digital health technology companies offering dementia preventive lifestyle services

The systematic search and rigorous selection process culminated in identifying 10 DDLS companies (see Table 1), collectively amassing EUR 128.52 million in funding. This remarkable funding concentration, predominantly within three companies (more than 75% of the total funding), underscores the competitive and uneven landscape of digital dementia prevention initiatives. The diversity in the years of establishment among these companies highlights an evolving field (i.e., 1999–2021), with both longstanding entities and new entrants driving innovation in dementia preventive services.

Our analysis revealed a significant European presence among the top-funded companies, alongside notable representations from North America and Asia, however we also consider our language exclusion criteria as a potential limitation – which we discuss further in this paper. This geographical distribution emphasizes the global interest in digital solutions for dementia prevention, transcending regional boundaries to address a universal public health challenge. Table 1 provides a summary of the 10 identified DDLS companies.

### Evidence analysis

The analysis of nine publications revealed a skewed distribution of evidence across three companies: Cognifit (five publications), Beynex, and Constant Therapy Health (two each). This indicates that 7 out of 10 identified DDLS provided no evidence meeting our criteria. No direct link was found between a company's funding and its number of relevant publications. Despite Constant Therapy Health's significant funding (EUR 29.3 million), it did not lead in publication count, while Beynex, with only EUR 0.1 million in funding, matched its output.

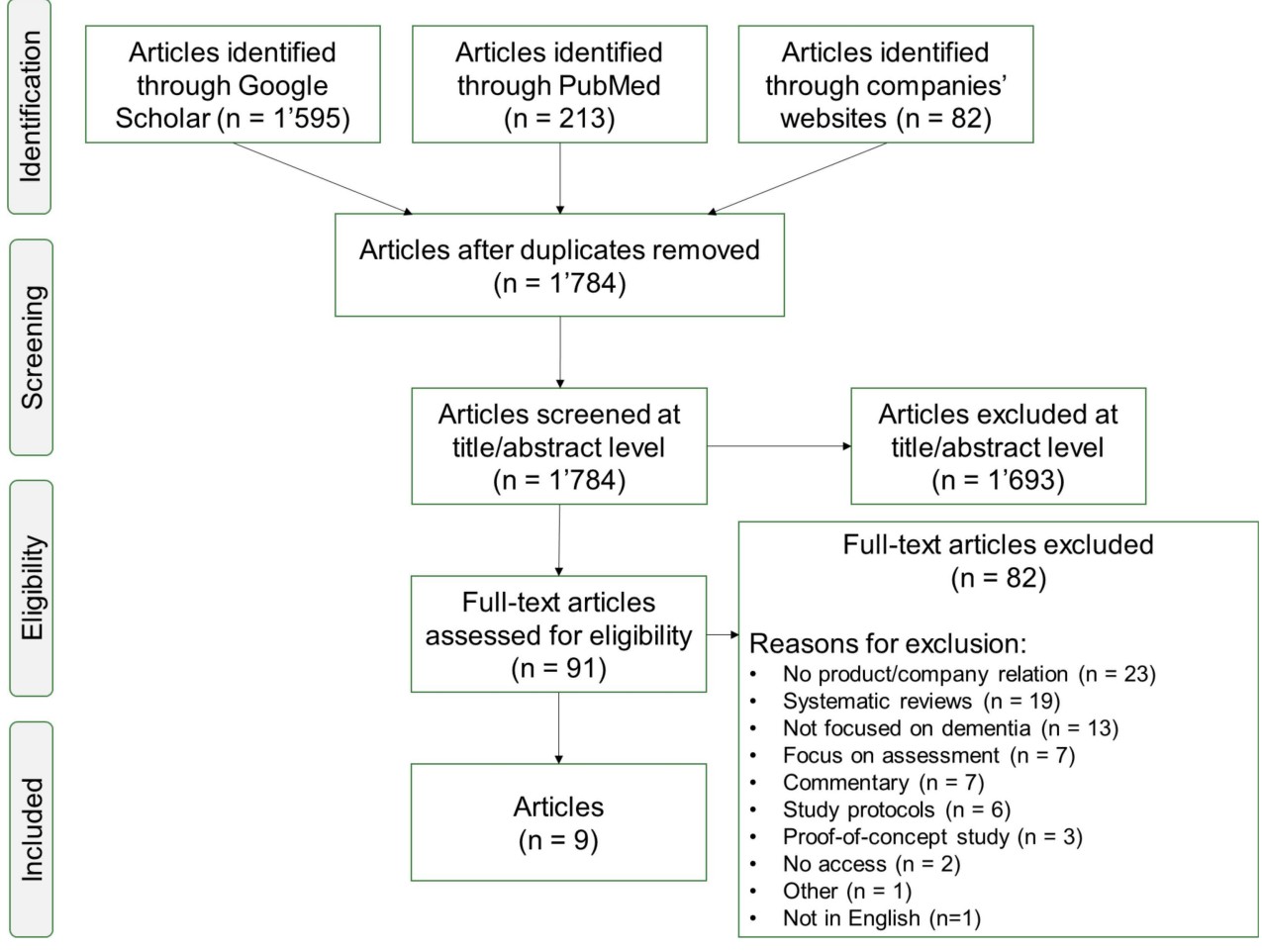

**Fig 2. Flow diagram for the included publications in the evaluation of clinical evidence.**

The publication dates of studies ranged from 2013–2022. Eight studies were published as papers in journals and one as a poster presentation at a conference. Five publications are indexed in PubMed and six are published in papers with an assigned journal impact score (JIS). With an average JIS of 4.29, the lowest journal had a JIS of 2.31 and the highest 6.591. Three of the studies were officially registered trials, and three others were IRB approved and/or a published study protocol.

Out of nine studies, five did not fit the Clinical Effectiveness or Validation categories as previously defined [26], including three feasibility studies and two comparing interventions in tailored vs. untailored settings. Three studies aimed at clinical outcomes, focusing on clinical outcomes, and one validated outcomes against another solution.

Of the nine studies assessed for USPSTF level of evidence (clinical effectiveness publications only), all measured impact in terms of patient outcomes, and none did so in terms of cost or access to care. Seven publications showed level 1 (good) evidence according to the criteria of the USPSTF, meaning that evidence was generated through at least one randomized-controlled trial. Out of all the publications, only one demonstrated level 2 (fair) evidence, while another publication presented level 3 (poor) evidence.

Average participation was n = 59, ranging from n = 2 to n = 122, with most studies (7 out of 9) involving n < 100 participants. Six of the studies targeted a study population with a condition and only two targeted participants with a disease risk

**Table 1. List of all identified solutions that correspond with the pre-defined set of inclusion/exclusion criteria. (Extracted company and funding data is per January 30, 2023).**

| # | Company | Year founded | Total number of employees | Total funding (mEUR) | Year of last funding |
|---|---------|--------------|---------------------------|----------------------|----------------------|
| 1 | Cognifit | 1999 | 55 | 6.21 | 2022 |
| 2 | NeuroNation | 2011 | 25 | 19.06 | 2021 |
| 3 | Neurotrack | 2012 | 41 | 50.21 | 2022 |
| 4 | Constant Therapy Health | 2013 | 19 | 29.35 | 2017 |
| 5 | MindStep | 2017 | 25 | 3.31 | -[b] |
| 6 | Luci | 2017 | 25 | >= 0.47[a] | -[b] |
| 7 | Five Lives | 2019 | 36 | 6.18 | 2022 |
| 8 | OptiChroniX | 2019 | 7 | 0.32 | 2022 |
| 9 | Beynex | 2020 | 6 | 0.1 | 2023 |
| 10 | Emogoc | 2021 | 30 | 13.31 | 2022 |
| | Average | 2015 | 27 | 12.85 | 2021 |

[a]Unable to retrieve total funding information, thus it is possible that the overall funding is higher.

[b]No information regarding last funding round available.

factor. The most common targeted condition was Subjective Mild Cognitive Impairment (SMCI) and Mild Cognitive Impairment (MCI) itself, followed by dementia, primarily encompassing Alzheimer's disease (AD). The two targeted risk factors included diabetes and old age. None of the studies demonstrated any change in the targeted condition or risk factor.

Except for 1 study, all studies demonstrated shifts in study proxies, mainly showing improved cognitive and memory functions (see Table 2). Overall, the quality of the analyzed publications was assessed as rather mediocre with an average subjective quality score of 3.

## Discussion

The aim of this work was to provide an initial mapping of the globally top-funded DDLS and to analyze the currently available body of published clinical evidence linked to these services. 10 companies with a total funding of EUR 128.52 million, headquartered in eight different countries have been identified. Funding ranged from EUR 0.1 million to EUR 50.21 million, with the top two companies accounting for over half of the total funding. No clear correlation between a company's founding year and its funding was found. Clinical evidence meeting our criteria was scarce, with only nine studies from three companies found. Many studies did not focus on clinical effectiveness or validation, three focused on feasibility, three on clinical effectiveness, and one on validation against alternatives. Most studies involved subjects with MCI or AD, with only two targeting subjects at risk. While 78% of the studies used randomized-controlled trials, sample sizes were small, and findings mainly showed changes in proxies rather than direct impacts on targeted conditions.

### Interpretation of results

**Companies offering DDLS.** The aging global population is making the social and economic impacts of dementia increasingly severe, with the WHO highlighting the urgency of prevention through lifestyle changes [7,15], and the provision of digital health interventions (DHIs) could significantly contribute to those efforts. Despite this, DHIs for dementia prevention seem underfunded compared to other areas, such as depression DHIs, which received more funding for the fifth-best funded initiative than all analyzed dementia companies combined [29]. The impression of low overall funding of DDLS companies is confirmed when considering the staggering USD 23,796 estimated global societal cost of dementia per person with dementia in 2019 [30].

**Table 2. List of all included publications that correspond with the pre-defined set of inclusion/exclusion criteria.**

| #[a] | Company | Reference | Purpose | # clinical subjects | Condition/ risk factor targeted | Change in incidence | Change in proxy | Level of evidence | Publication quality |
|---|---|---|---|---|---|---|---|---|---|
| 1 | Beynex | [29] | Clinical effectiveness | 120 | • AD<br>• Subjective memory complaint | No | Improved MoCA scores Bayer-ADL scores indicated improvement in ADL. | 1 | 1 |
| 2 | Beynex | [30] | Clinical effectiveness | 60 | Subjective CI | No | Improved memory related cognitive parameters. | 1 | 1 |
| 3 | Cognifit | [31] | Other (Tailored/ untailored setting for subjects & self-efficacy/ no self-efficacy) | 84 | Diabetes | No | Improved global cognition and memory composite scores. | 1 | 4 |
| 4 | Cognifit | [32] | Other (Tailored/ untailored setting for subjects) | 44 | • MCI<br>• MrNPS | No | Improved performance on composite measures of global cognition, learning, delayed epi-sodic memory. | 1 | 4 |
| 5 | Cognifit | [33] | Validation | 47 | CI | No | Improved performance of global cognition, working memory, divided attention, pro-cessing speed. | 1 | 4 |
| 6 | Cognifit | [34] | Other (feasibility combined with effectiveness) | 18 | • Episodic memory dysfunction<br>• MCI | No | Improved working memory and speed. | 2 | 3 |
| 7 | Cognifit | [35] | Clinical effectiveness | 122 | Old age | No | Improved visual-spatial information processing, visual scanning, global visual memory, naming, hand-eye coordination, visuospatial learning, and visuospatial work-ing memory. | 1 | 5 |
| 8 | Constant Therapy | [36] | Other (feasibility) | 2 | Dementia | No | No. | 3 | 2 |
| 9 | Constant Therapy | [37] | Other (feasibility) | 19 | AD | No | Improved visual & audi-tory memory, attention, arithmetic, processing speed, adaptability. | 1 | 3 |
| Average | | | | 59 | | | | | 3 |

AD, Alzheimer's disease; ADL, activities of daily living; CI, cognitive impairment; MCI, mild cognitive impairment; MoCA, Montreal cognitive assessment; MrNPS, mood-related neuropsychiatric symptoms.

[a]Different coloring represents company affiliation of the study.

Since the underlying reasons for those findings are beyond the scope of this study, it should rather briefly be touched upon potential causes: A potential explanation are the relatively young medical findings that set the foundation for the offered interventions. A landmark study showed the positive impact of a multi-domain lifestyle intervention on dementia risk in 2015 and thus is not much older than a major part of the identified companies [13]. On the other side, however, many of the identified companies strongly leverage brain games as a way to foster cognitive engagement. The idea of

utilizing cognitive training to prevent or delay dementia has been widely discussed before [31–34]. The companies' recent establishment may explain the modest funding levels, challenging the assumption that funding correlates with company age. Monetization uncertainties and the prevalence of local champions, suggesting a fragmented market with limited global commercial potential, could also impact funding. These hypotheses underscore the need for further research in this area.

**Clinical evidence.** The limited number of publications meeting our criteria, primarily from just three companies, is unexpected. This is particularly striking considering the companies' professed strong scientific orientation and their extensive citation of scientific papers on their websites [35–40]. In addition to the well-documented challenges of applying traditional evidence generation methods to digital health interventions [41], previous research suggests that younger companies are less likely to have completed and published full clinical trials, as evidence generation tends to lag behind product launch in fast-moving digital health markets [42]. This time lag between founding and the availability of peer-reviewed evidence could partially explain the low number of eligible publications identified in this study. A similar disconnect between scientific positioning and published evidence has also been observed in other areas of digital health [29,42].

The prominence of companies focusing on cognitive engagement through brain exercises (Constant Therapy Health, Cognifit, Beynex) in our findings is notable. Brain training has been a significant research focus, making it more probable for these solutions to have accumulated evidence. Cognifit, the oldest company (founded in 1999), leads with five publications, including two from 2013 [43]. However, the newer companies, founded in 2013 (i.e., Constant Therapy Health) and 2020 (i.e., Beynex), along with the more recent average publication year of 2020 for remaining studies, suggest another reason for this trend. By requiring subjects to perform only game-based brain training with relatively little effort (compared to changing diet and exercise habits), drop-out rates could be lower, adherence rates higher, the trial more manageable and short-term effects potentially faster to detect. Thus, the failure rate of those studies is lower [44] and the overall risk due to extensive existing research diminished.

The evidence quality from the limited publications is mediocre, with most studies using randomized-controlled trials (evidence level 1 according to previously determined research [26]), but involving small groups and showing changes only in utilized proxies. Solutions aimed at reducing the risk of MCI progressing to AD targeted subjects with MCI but only showed improved cognitive performance, not prevention of conversion. A potential reason for this could be the additional financial and non-financial resources necessary to conduct longitudinal studies at a large scale. Most of the identified studies did not measure clinical endpoints or dementia incidence directly, but instead reported changes in proxies, such as cognitive performance [45–53] or memory function [13,52,53]. Such proxies are commonly used in dementia prevention research, particularly in short-term trials where observing disease onset would not be feasible [54]. While these proxy improvements do not constitute direct evidence of reduced dementia incidence, they are aligned with established mechanisms linking cognitive reserve and lifestyle factors to lower dementia risk [55,56].

Notably, none of the identified studies explicitly measured the impact of DDLS on cost or access to care, despite these being key evaluation criteria in previous digital health market analyses [26]. While clinical effectiveness remains central to prevention, improving access to such interventions is also a critical enabler of their impact. This absence reflects a broader gap in research on how digital solutions enhance accessibility to dementia-preventive interventions, particularly for at-risk populations who face barriers to traditional in-person care, such as geographic limitations, socioeconomic constraints, or healthcare provider shortages. The low number of scientific studies and identified methodological issues are in line with previous findings [57], where researchers systematically analyzed clinical evidence of mobile health solutions for people suffering from dementia and their relatives. On this basis, it was concluded that there is no evidence for the clinical effectiveness of the analyzed services [57]. Considering the few publications and their methodological limitations, this work suggests an insufficiency of evidence for the effectiveness of top funded DDLS internationally.

## Theoretical contributions

Overall, the results align with the complex definitions found in existing literature. Despite clear theoretical guidelines for analysis, comparing services highlights blurred distinctions between terms. Notably, a managerial aspect supports clinicians with non-clinical tasks, such as documentation, following the Digital Medicine Society (DiMe)'s framework [58]. This indicates that digital health components can be part of digital medicine offerings, challenging the clear differentiation suggested by initial terminology. This relates to strategies like those of Constant Therapy Health and Cognifit, which aim to integrate into existing care processes rather than just complement them, raising questions about the adequacy of assumed definitions and the potential need for new concepts.

The foundation of DHIs is notably their evidence base. The results of the systematic review found that evidence supporting identified DDLS is scarce, with many companies focusing on general reviews rather than assessing their products' clinical effectiveness or comparing them to other interventions. This highlights a need for more precise definitions within DHI and Digital Medicine fields. While it is challenging to reclassify these solutions as merely lifestyle or wellness apps without medical relevance, the current definitional framework lacks specificity. A more holistic approach to classification is suggested, one that not only evaluates evidence but also considers business models, offering a broader perspective on DHIs' role and impact. In addition to the previously mentioned benefits of clarity of definitions, this could also further strengthen the bridge between the role of evidence generation and business model building and scaling in digital health: When digital medicine companies operate at the crossroads between regular technology companies and pharmaceutical companies, obtaining clinical evidence is paramount [59].

## Clinical implications

The current manuscript undertakes a critical exploration into the realm of DDLS, scrutinizing the clinical evidence that underpins these emerging interventions. This inquiry is paramount, not only due to the growing investments and interest in DHIs aimed at staving off dementia but also because it addresses a significant gap in existing literature. The pressing need for effective dementia prevention strategies, in the absence of disease-modifying treatments, underscores the importance of this study.

Evaluating the relevance and novelty of this research, it becomes evident that it fills an essential void by systematically identifying and analyzing the top funded companies offering DDLS. This approach not only sheds light on the current landscape of digital interventions but also critically assesses the extent and quality of clinical evidence supporting their efficacy. In doing so, the study brings forth new perspectives on the role of digital health in preventing dementia, challenging existing paradigms by questioning the robustness of the purported benefits of these interventions.

The study's findings on the limited clinical evidence supporting the efficacy of DDLS highlight the urgent need for more rigorous and longitudinal research in this area. Such evidence is crucial for informing clinical guidelines, shaping public health policies, and guiding future research directions. The identification of this gap not only signals the necessity for further empirical inquiry but also posits the manuscript as a cornerstone for subsequent investigations aimed at validating and enhancing the clinical utility of digital interventions for dementia prevention.

Moreover, the interdisciplinary nature of the manuscript, which intersects medical science, digital technology, and health policy, exemplifies the complex and multifaceted approach required to tackle dementia prevention. The manuscript's exploration of the funding dynamics and the technological underpinnings of DDLS, coupled with its analysis of clinical evidence, reflects a comprehensive understanding of the ecosystem surrounding DHIs for dementia. This interdisciplinary perspective is vital for devising holistic and effective prevention strategies that can be seamlessly integrated into public health frameworks and clinical practice.

The manuscript also acknowledges the paramount importance of patient and public involvement in the research and development of DDLS. This recognition aligns with contemporary research ethics, emphasizing the co-creation of health interventions that are not only scientifically sound but also resonate with the needs, preferences, and realities of those

they aim to serve. Such an approach not only enhances the relevance and applicability of research findings but also ensures that digital health interventions are grounded in the lived experiences of individuals at risk of dementia, thereby maximizing their potential impact.

Lastly, the manuscript's call for transparency and availability of data is a testament to its commitment to the principles of open science. By advocating for the unrestricted sharing of research data and methodologies, the study sets a standard for future research in the field, facilitating the replication and validation of findings, and fostering a collaborative research environment that accelerates the advancement of knowledge and the development of effective dementia prevention strategies.

In summary, the clinical implications of this manuscript extend beyond the mere analysis of current DDLS. It lays the groundwork for future research, encourages interdisciplinary collaboration, and underscores the importance of patient and public involvement in the creation of DHIs. Furthermore, it champions the principles of transparency and open science, essential for the robust, ethical, and impactful advancement of dementia prevention research.

## Limitations

The company search (Study 1) has two major limitations: As the search was limited to companies with available information in English, the sample may underrepresent companies operating primarily in non-English-speaking markets, despite Pitchbook and Crunchbase both being global financial databases. In total, three companies were excluded due to lacking English-language information. While the impact on the final sample was minor, this language bias should be considered when interpreting the findings as globally representative.

Additionally, only companies with publicly available funding information were analyzed, possibly omitting other significant players. Data completeness also affects the accuracy of funding-based rankings, as market intelligence databases may not have full funding details. The publication analysis (Study 2) has further limitations in relation to our second research question. Firstly, while Google Scholar search results may include the full text of open access publications, PubMed searches only covered structured fields such as titles, abstracts, author lists, and (for recent publications) author affiliations. As a result, the identification of relevant publications primarily depended on the explicit mention of company or solution names in these searchable fields. This approach may have missed studies conducted by independent academic partners or published prior to product commercialization, where company or product names might be less likely to be referenced. However, for the top-funded companies targeted in this study, we consider the likelihood of explicit product or company name inclusion to be relatively high, as digital health companies typically have a strong commercial interest in showcasing clinical evidence to support regulatory approval, payer negotiations, or market positioning—particularly in regulated health markets [60,61].

Secondly, in some cases, search results were narrowed using the term "Dementia" to focus on DDLS in alignment with our research objective. While necessary for refining search relevance, this approach may have excluded studies examining broader cognitive health interventions that could have indirect relevance to dementia prevention but do not explicitly mention dementia in their title, abstract, or keywords.

Thirdly, strict inclusion and exclusion criteria meant that studies on assessment/or diagnostic tools and those not focused on dementia prevention were excluded, possibly indicating a lack of clinical evidence for the companies reviewed.

Fourthly, while study duration was considered, the specific length of interventions wasn't, leaving some potential explanations unexplored.

Fifthly, our analysis did not compare the clinical effectiveness of the solutions directly but evaluated the quality and results of each study independently, without comparing them to one another.

## Suggestions for future research

Despite the promising strides in developing and funding DDLS, the study highlighted a critical gap in the clinical evidence underpinning these interventions. The limited scope of published studies, small participant groups, and the absence of longitudinal research point to an emergent field still grappling with establishing a robust evidence base.

The disconnect between the proliferation of funded initiatives and the paucity of rigorous clinical validation underscores the nascent stage of digital interventions in dementia prevention, marking a crucial area for future research and development.

Avenues for further research also include the screening of the global landscape with a focus on local champions, since the identification process showed that there are several highly interesting solutions which are only offered in a local language and setting. This research may open new possibilities for studying the blending of lifestyle interventions into local surroundings like hiking areas (physical activity) or community clubs (cognitive engagement). This could significantly contribute to the development of best practice reference models in the field of clinical evaluation itself as well as company building and business model development as a whole.

## Supporting information

**S1 Table. Search categories and keywords for the Pitchbook search.**
(DOCX)

**S2 Table. Search categories and keywords for the Crunchbase search.**
(DOCX)

## Acknowledgments

A substantial part of work has been conducted by JP as part of his master's thesis at the University of St. Gallen.

## Author contributions

**Conceptualization:** Rasita Vinay, Tobias Kowatsch, Marcia Nißen.

**Data curation:** Jonas Probst.

**Formal analysis:** Jonas Probst.

**Investigation:** Jonas Probst.

**Methodology:** Rasita Vinay, Tobias Kowatsch, Marcia Nißen.

**Project administration:** Rasita Vinay, Jonas Probst, Tobias Kowatsch, Marcia Nißen.

**Resources:** Rasita Vinay, Marcia Nißen.

**Supervision:** Rasita Vinay, Tobias Kowatsch, Marcia Nißen.

**Validation:** Rasita Vinay, Marcia Nißen.

**Visualization:** Rasita Vinay, Marcia Nißen.

**Writing – original draft:** Rasita Vinay, Jonas Probst, Panitda Huynh, Mathias Schlögl, Tobias Kowatsch, Marcia Nißen.

**Writing – review & editing:** Rasita Vinay, Marcia Nißen.

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
