## [Decision Letter · Decision Letter 0]

13 Sep 2024

PONE-D-24-16712Top-funded digital health companies offering lifestyle interventions for dementia prevention: Company overview and evidence analysisPLOS ONE

Dear Dr. Vinay,

Thank you for submitting your manuscript to PLOS ONE. After careful consideration, we feel that it has merit but does not fully meet PLOS ONE’s publication criteria as it currently stands. Therefore, we invite you to submit a revised version of the manuscript that addresses the points raised during the review process.

We look forward to receiving your revised manuscript.

Kind regards,

Rocktotpal Konwarh, Ph.D.

Academic Editor

PLOS ONE

Journal Requirements:

2. Thank you for stating the following in the Competing Interests section: “RV, PH, TK, and MN are affiliated with the Centre for Digital Health Interventions (CDHI), a joint initiative of the Institute for Implementation Science in Health Care, University of Zurich, the Department of Management, Technology, and Economics at the ETH Zurich, and the institute of Technology Management and School of Medicine at the University of St. Gallen. CDHI is funded in part by CSS, a Swiss health insurer, Mavie Next, an Austrian healthcare provider and MTIP, a Swiss investor company. TK is also a co-founder of Pathmate Technologies, a university spin-off company that creates and delivers digital clinical pathways. However, neither CSS nor Pathmate Technologies, Mavie Next, or MTIP was involved in this research. All other authors declare no conflict of interest.”

 Please confirm that this does not alter your adherence to all PLOS ONE policies on sharing data and materials, by including the following statement: "This does not alter our adherence to PLOS ONE policies on sharing data and materials.” (as detailed online in our guide for authors http://journals.plos.org/plosone/s/competing-interests). If there are restrictions on sharing of data and/or materials, please state these. Please note that we cannot proceed with consideration of your article until this information has been declared. Please include your updated Competing Interests statement in your cover letter; we will change the online submission form on your behalf.

3. We note that your Data Availability Statement is currently as follows: “All relevant data are within the manuscript and its Supporting Information files.”

Please confirm at this time whether or not your submission contains all raw data required to replicate the results of your study. Authors must share the “minimal data set” for their submission. PLOS defines the minimal data set to consist of the data required to replicate all study findings reported in the article, as well as related metadata and methods (https://journals.plos.org/plosone/s/data-availability#loc-minimal-data-set-definition). For example, authors should submit the following data: - The values behind the means, standard deviations and other measures reported; - The values used to build graphs; - The points extracted from images for analysis. Authors do not need to submit their entire data set if only a portion of the data was used in the reported study. If your submission does not contain these data, please either upload them as Supporting Information files or deposit them to a stable, public repository and provide us with the relevant URLs, DOIs, or accession numbers. For a list of recommended repositories, please see https://journals.plos.org/plosone/s/recommended-repositories. If there are ethical or legal restrictions on sharing a de-identified data set, please explain them in detail (e.g., data contain potentially sensitive information, data are owned by a third-party organization, etc.) and who has imposed them (e.g., an ethics committee). Please also provide contact information for a data access committee, ethics committee, or other institutional body to which data requests may be sent. If data are owned by a third party, please indicate how others may request data access.

Reviewers' comments:

Reviewer's Responses to Questions

**Comments to the Author**

1. Is the manuscript technically sound, and do the data support the conclusions?

Reviewer #1: Partly

Reviewer #2: Yes

2. Has the statistical analysis been performed appropriately and rigorously? 

Reviewer #1: Yes

Reviewer #2: Yes

3. Have the authors made all data underlying the findings in their manuscript fully available?

Reviewer #1: Yes

Reviewer #2: Yes

4. Is the manuscript presented in an intelligible fashion and written in standard English?

Reviewer #1: No

Reviewer #2: Yes

5. Review Comments to the Author

Reviewer #1: • More clarification of the term “invisible second patients” needed.

• No effective treatment for dementia management is available? (Cite the specific literature, if available, for this particular statement. There are so many pharmacological and non-pharmacological approaches to manage dementia)

• Methodology section is too limited. More clear explanation is needed.

• In Table No.1 the founding year of each company could have been arranged sequentially.

• From line 261-263 along with level try to provide the category name like – level 1 – good, level 2 – fair and so on.

• 268 – it is commonly known to everyone that, people with old age suffer from cognitive impairment (dementia). The outcome(s) of the review should be novel.

• 319- The lack of publication may be due to the young age of the companies. Can personal assumption be made in a systemic review or in any kind of scientific review???

• The paper somewhere failed to highlight the digital initiative in managing dementia or reducing the risk of dementia. Moreover, the improvements were just mentioned in a column under the heading “Change in Proxy”.

• The advantage of digital management as compared to in-person cognitive training could have been mentioned. Moreover, even the benefits of DHI’s and how it challenged the existing paradigms remain unclear.

• Research evidences pertaining to this topic are scarce, still the current paper claims to fill the existing literature gap.

• The sole aim of the paper somewhere got lost. The paper has lost its track from its central objective. Unnecessary details have been mentioned.

• Finally, the scope of the review was found to be scant.

Reviewer #2: PLOS One Manuscript review 08/09/24

Top-funded digital health companies offering lifestyle interventions for dementia prevention

Suggestion: Major revisions

Major comments

Line 133 Data was collected up to January 2023. Consider updating the search to include the last 20 months?

Line 176 The exclusion of non-risk groups (healthy individuals) seems to contradict the focus on prevention. Surely the target prevention population is dementia-free individuals?

Minor comments

Line 36 “Lack of disease modifying treatments” is no longer accurate. Suggest rewording.

Line 52 The conclusion is quite vague and lacks specific direction, suggest rewording to make clear what the added value of this study is.

Line 57 Avoid using the term “suffer” to refer to people living with dementia.

Make clear what is meant by dementia. Alzheimer’s disease? Or all-cause dementia syndromes? Those with a clinical diagnosis? Pathology?

Line 74 “Lack of effective treatments” no longer accurate.

Line 76 make clear what stage of life is being referred to when you talk about lifestyle. The potential impact of lifestyle changes is not constant across the lifespan.

Line 78 Cognitive training is not a well established modifiable risk factor for dementia, so it seems a little odd to include this here, particularly when other risk factors with better evidence are not mentioned.

The phrase ‘directly influence’ is problematic as the evidence is largely not causal and many interacting factors are involved.

Reference 10 is out of date, suggest replacing with an updated figure from more recent research.

Line 98 Suggest specifying ‘published’ clinical evidence – as you’ve not reviewed clinical evidence outside of the academic literature.

Line 99 I didn’t follow this. What is being continuously peer reviewed?

Line 104 Was the Google search systematic? What was the reason to do this?

Line 110 Search terms seem very limited. E.g were Alzheimer*, brain health etc not considered?

Line 121 The abstract suggests this study identifies companies globally, however the method section here excluded solutions not written in English. Surely this will limit the ability to accurately capture relevant solutions globally? Suggest removing the global claims given the methodology, and acknowledging this limitation in the discussion section.

Line 145 Please outline all reasons for exclusion with the number excluded for each reason. A flowchart may be helpful to illustrate how you reached the final analytical sample from the initial pool.

Line 166 The reliance on the company/ solution name being included in the publication is a limitation of this study which should be acknowledged in the limitations section. The company or solution name will not necessarily be included if the study was conducted by an academic partner, or prior to the solution being marketed. Also please state in the methods whether the company/solution name searched across the title/ abtract? Or author list as well?

As company websites were reviewed, did you consider contacting the companies to ask for signposting to their clinical evidence?

Line 171 Again, dementia seems rather limited as a search term on it’s own.

Line 174 – Why refer to access to care here? How is this related to prevention?

Line 221 Geographical analysis – I’m not sure what conclusions can be drawn from these results given the language restriction used in the methodology, as mentioned earlier.

Line 230 This whole paragraph seems to stray away from scientific results from the data collected. Suggest reducing rhetoric not related to the data, and saving interpretation for the discussion section.

6. PLOS authors have the option to publish the peer review history of their article (what does this mean? ). If published, this will include your full peer review and any attached files.

**Do you want your identity to be public for this peer review?** For information about this choice, including consent withdrawal, please see our Privacy Policy .

Reviewer #1: No

Reviewer #2: No

---

## [Author Response · Author response to Decision Letter 1]

7 Mar 2025

Dear Dr. Konwarh,

We are grateful for the invitation to revise our manuscript “Top-funded digital health companies offering lifestyle interventions for dementia prevention: Company overview and evidence analysis” [PONE-D-24-16712]. We appreciate your and the reviewers’ constructive and knowledgeable feedback, which we have incorporated into our revised manuscript by attending to all comments carefully. We also want to thank you for granting us an extension for resubmission, as we were not able to get the whole team together for the original resubmission deadline in October 2024. It is kindly appreciated.

Please find our detailed answers to the reviewer comments as an attached file. Overall, we sincerely thank the reviewers for their helpful thoughts, comments, advice, and valuable concrete suggestions for improvement. We hope that the revision of our manuscript substantially advances the paper to move it forward in the process.

We are looking forward to hearing from you and will respond to any further questions and comments that you may have.

Sincerely,

<anonymized>

---

## [Decision Letter · Decision Letter 1]

8 Apr 2025

Top-funded digital health companies offering lifestyle interventions for dementia prevention: Company overview and evidence analysis

PONE-D-24-16712R1

Dear Dr. Rasita Vinay,

We’re pleased to inform you that your manuscript has been judged scientifically suitable for publication and will be formally accepted for publication once it meets all outstanding technical requirements.

Kind regards,

Rocktotpal Konwarh, Ph.D.

Academic Editor

PLOS ONE

---

## [Editor Report · Acceptance letter]

PONE-D-24-16712R1

PLOS ONE

Dear Dr. Vinay,

I'm pleased to inform you that your manuscript has been deemed suitable for publication in PLOS ONE. Congratulations! Your manuscript is now being handed over to our production team.

Kind regards,

on behalf of

Dr. Rocktotpal Konwarh

Academic Editor

PLOS ONE